# Bovine Immunity and Vitamin D_3_: An Emerging Association in Johne’s Disease

**DOI:** 10.3390/microorganisms10091865

**Published:** 2022-09-18

**Authors:** Taylor L. T. Wherry, Judith R. Stabel

**Affiliations:** 1Department of Veterinary Pathology, Iowa State University, Ames, IA 50011, USA; 2Infectious Bacterial Diseases Research Unit, United States Department of Agriculture-Agricultural Research Service (USDA-ARS), National Animal Disease Center, Ames, IA 50010, USA

**Keywords:** vitamin D, *Mycobacterium avium* subsp. *paratuberculosis*, cattle, macrophage, PBMC, endosomal trafficking

## Abstract

*Mycobacterium avium* subspecies *paratuberculosis* (MAP) is an environmentally hardy pathogen of ruminants that plagues the dairy industry. Hallmark clinical symptoms include granulomatous enteritis, watery diarrhea, and significant loss of body condition. Transition from subclinical to clinical infection is a dynamic process led by MAP which resides in host macrophages. Clinical stage disease is accompanied by dysfunctional immune responses and a reduction in circulating vitamin D_3_. The immunomodulatory role of vitamin D_3_ in infectious disease has been well established in humans, particularly in *Mycobacterium tuberculosis* infection. However, significant species differences exist between the immune system of humans and bovines, including effects induced by vitamin D_3_. This fact highlights the need for continued study of the relationship between vitamin D_3_ and bovine immunity, especially during different stages of paratuberculosis.

## 1. Johne’s Disease Overview

Johne’s disease, also called paratuberculosis, is a chronic, progressive enteritis affecting ruminants and has economic importance particularly within the dairy industry. The etiological agent, *Mycobacterium avium* subspecies *paratuberculosis* (MAP), is an acid-fast bacillus that is classified as an obligate intracellular pathogen [1]. Estimates of MAP’s impact on economic losses have varied through the years, with the most recent values reaching $198 million dollars per year in the United States alone [2,3].

Persistence of MAP within the environment is achieved as a result of its thick, waxy cell wall composed of long mycolic acids. These fatty acids provide a uniquely impermeable barrier that protects the bacterium from desiccation and damage by chemical or biological agents [4,5]. Pasture-based grazing systems face unique challenges in the control of paratuberculosis, as MAP can be isolated from grass and top layers of soil following manure application [6]. Environmental conditions have been shown to impact MAP longevity with dry, shaded areas of land testing positive up to 55 weeks [7] and water also potentially being a prominent environmental reservoir, testing positive for up to 36 weeks [8]. Other barriers to eradication and control include the potential for a wildlife reservoir, such as deer, that can continue the disease cycle and infect domesticated livestock [9,10].

## 2. Transmission

MAP infection rates in young calves increase with exposure to larger numbers of MAP, and infection is most commonly established via the fecal–oral route for upwards of a year following birth [11,12]. Transmission of MAP through colostrum and milk by infected dams has been shown to occur and detection rates, along with bacterial burden, appear to coincide with the level of fecal shedding by the dam [13]. Further, rates of *in utero* transmission also appear to relate to disease severity and MAP burden in the feces [14,15]. These routes of infection represent important steps in biosafety protocol during efforts of herd infection control, and largely revolve around maintaining clean, dry housing environments.

Fecal shedding patterns are intermittent in animals naturally infected with MAP categorized in the subclinical stage of disease. Dynamics of fecal shedding have been shown to be largely different between experimentally and naturally infected cattle, with experimentally infected animals shedding after one year and alternating between positive and negative status several times while also fluctuating between high and low levels [16]. Most of these animals reached high levels of shedding, which could be impacted by MAP infection dose and administration route. This study also showed those animals that were naturally infected at a young age had a propensity to begin shedding approximately 3 years following infection, had little to no switching between shedding status, and those that did shed intermittently had a reduced chance of becoming a high shedder compared to those who shed MAP consistently. Other reports have also shown similar disease outcomes related to shedding level and frequency [17]. Additionally, naturally infected cattle that began shedding high levels frequently continued to do so until being removed from the herd [16].

Cattle that progress to clinical stage paratuberculosis show symptoms including watery diarrhea, weight loss, and in severe cases submandibular edema. Gross pathology shows marked inflammation of the intestinal wall, which impairs nutrient absorption from the diet [18]. This can lead to hypoproteinemia, reducing oncotic pressure of capillary vessels allowing fluid to leak from the blood vessels and accumulate beneath the jaw in grazing animals. The clinical stage is often accompanied by shedding of high levels of MAP and a significant reduction in milk production for lactating animals [19].

Furthermore, MAP has been implicated in Crohn’s disease in humans, although this is an ongoing controversial relationship [20]. Association has been established through studies isolating MAP genomic DNA from Crohn’s patients, although inconsistently, and some individuals have experienced remission of symptoms following antimycobacterial treatments [21,22,23]. The plausibility of a relationship is further supported by MAP’s ability to survive pasteurization of dairy products [24]. Collectively, a definitive causal link has yet to be established and considerations should be made as to whether MAP is simply an opportunistic pathogen in humans that is able to establish itself following disease onset.

## 3. Detection and Diagnostics

One main goal of on-farm paratuberculosis control programs includes identifying potential carriers and removing them from the herd. Unfortunately, a practical and consistent diagnostic tool does not yet exist for routine implementation by veterinarians and producers. Challenges in diagnosing early infection include the sensitivity of antibody enzyme-linked immunosorbent assay (ELISA) and fecal culture, which has been shown to increase with host age and can therefore better detect infection as it advances [25]. Cattle mounting a cell mediated immune response will produce significant amounts of antigen-specific IFN-γ, which can be detected by ELISA [26,27]. Additionally, detection of MAP genomic material in host samples utilizing qPCR tools is commonly achieved by targeting a species-specific insertion sequence (IS*900*), of which there are approximately 16–22 copies [28], making this a sensitive diagnostic tool.

As development of paratuberculosis is not always linear, different methods of diagnostics do not always have a high rate of outcome agreement. For example, anti-MAP antibody production is not a great predictor of fecal shedding, and disparate results among antibody production, fecal culture, and fecal PCR can occur [29]. Additionally, postmortem histopathologic examination of host tissues can be less sensitive, so more accurate detection of infected animals can be achieved by performing another concurrent diagnostic such as culture of homogenized tissues [30]. To address the issue of time intensive traditional methods of culture on solid media, culture of samples in liquid media, such as Middlebrook 7H9, allows for a reduced window of detection and more rapid removal of the infected animal from the herd during the dry period [31,32,33].

## 4. Host Entry and Recognition

To establish infection via the fecal-oral route, MAP must traverse the mucosal epithelium in the small intestine. It is here where highly specialized areas of gut-associated lymphoid tissue (GALT) are located, called Peyer’s Patches, which function in continuous sampling of the gut environment for potential pathogens. Microfold cells (M cells) located in Peyer’s Patches are gatekeepers for this surveillance role and MAP utilizes them as a doorway to reach antigen presenting cells in the submucosa [34]. While the general consensus appears that M cells are the main route of entry into the host subepithelial tissue, MAP does facilitate entry into the host through intestinal epithelial cells, as reported in a Peyer’s Patch deficient mouse model [35] and an *in vivo* lamb small intestine surgical loop model [36].

Models incorporating simulations of MAP interaction with host epithelial cells have shown alterations in MAP gene expression and patterns of host cytokine secretion possibly because of phenotypic changes acquired by the bacteria. Pre-processing of MAP through Madin-Darby bovine kidney (MDBK) epithelial cells and its serial passage through RAW264.7 macrophages and MDBK cells showed secondary epithelial infection collectively increased gene expression of pro-inflammatory mediators interleukin 6 (IL-6) and IL-8 while also reducing anti-inflammatory TGF-β transcripts [37]. This study also showed gene expression patterns for pro-inflammatory CCL5, also known as Regulated Upon Activation, Normal T cell Expressed and Secreted (RANTES), and IL-18 were inversely related between primary and secondary epithelial cell cultures, with gene expression being upregulated in the secondary cell culture. Other work has observed increased expression of IL-17A and IL-17F following epithelial processing of MAP [38].

Successful establishment of infection is achieved following MAP uptake by antigen presenting cells, mainly subepithelial macrophages. The most well documented route of entry is facilitated through the cell surface pattern recognition mannose receptor binding MAP cell wall component mannose-capped lipoarabinomannan (ManLAM). Other mechanisms of entry have been identified including CD14, complement receptors CD11b/CD18, and integrin receptors CD11/CD18 [39]. Additionally, complement receptors CR1, CR3 and CR4 can facilitate uptake of mycobacteria opsonized with serum complement protein C3b [40]. Fc immunoglobulin receptor (FcR) recognizes and binds the Fc fragment of host antibody bound to the mycobacterium. Serum from MAP infected cattle used to opsonize MAP results in higher uptake of bacteria and the effect is maintained even when serum is heat inactivated [41]. In contrast, serum from naïve cattle that is heat inactivated does not increase bacterial uptake compared to neat serum. Toll-like receptors (TLRs) are a class of highly conserved transmembrane receptors that innately recognize pathogen associated molecule patterns (PAMPs) and damage associated molecular patterns (DAMPs) associated with a variety of pathogens and their products. TLR9 is thought to bind mycobacterial DNA within the cell, and TLR2 recognizes cell wall lipoproteins [42,43,44]. Each of these methods of macrophage signaling can have varying effects on its responses, and another influencing factor in that response is previous cellular activation.

## 5. Immune Responses to MAP

Macrophage phenotype and resulting effector functions are influenced by the cell’s cytokine milieu, pathogen signaling, and immune cell interactions. Phenotypic differences can be broadly represented by two groups according to cell surface receptors and cytokines produced [45]. A pro-inflammatory T helper 1 (Th1) cytokine profile, namely including IFN-γ, promotes a host defense (M1) macrophage phenotype with bactericidal activity that contributes to pathogen elimination. These cells have been found to produce high levels of nitric oxide and express high levels of CD80 and CD86 [46,47]. Binding of CD40 on the macrophage with CD40L (CD154) on activated T cells is essential for T cell proliferation and induction of pro-inflammatory responses including IL-12 expression and nitric oxide production [48,49,50]. In bovine monocyte-derived macrophages (MDMs) infected with MAP, defects in CD40-CD40L signaling resulted in abrogated gene expression for IL-12p40 and inducible nitric oxide synthase [51]. In contrast, uninfected MDMs were able to upregulate these responses in addition to increased IL-6, TNF-α, IL-8, and IL-10. The subclinical stage of disease is often associated with greater capacity to produce pro-inflammatory cytokines such as IFN-γ and TNF-α [26,27]. In the first year of infection with MAP, it has been shown CD4+CD45RO+ memory T cells are the main antigen responders, and significantly upregulate activation markers CD25 and CD26 following *in vitro* stimulation [52]. This study also observed that while CD8+CD45RO+ comprised approximately 30% of CD8+ T cells in the first year, they did not begin to have significant responses to *in vitro* stimulation until 18 months post infection, at which time they increased their expression of activation markers CD25 and CD26 [52].

Importantly, prior activation of the macrophage can have a profound effect on intracellular viability of the invading bacterium, which has been shown in IFN-γ pretreatment of macrophage *in vitro* [53,54]. Stimulation with a pro-inflammatory mediator after infection, however, does not confer the same protective effects. Furthermore, physiological responses such as polarization and signaling of T cells following antigen presentation rely on a variety of factors including antigen dose, its affinity for the T cell receptor (TCR), duration of binding, and production of anti-inflammatory IL-4 or pro-inflammatory IFN-γ [55,56].

The resolution and repair (M2) macrophage facilitates an environment in which a T helper 2 (Th2) cytokine profile predominates, defined by an anti-inflammatory response with increased IL-10 along with high levels of CD163 [57,58,59]. In the case of bovine paratuberculosis, the greater proportion of M2 macrophages present in the intestinal tissue of cows in clinical stage of disease ultimately hinders MAP clearance and disease resolution within the host [60,61]. This stage is characterized by a reduction in IFN-γ production, a cytokine which is protective for the host and functions by activating macrophages [27]. While the specific mechanisms are unresolved in this transition of immune function from subclinical to clinical cows, it has been shown that CD4+CD25- naïve T cells are largely unresponsive to MAP antigen and do not develop a regulatory T cell cytokine profile [62]. MAP-induced Th1 cytokine gene expression in these CD4+CD25- T cells was observed to be reduced in subclinical cows as well, but to a lesser extent. Furthermore, IL-10 facilitates reduction in IFN-γ and IL-12 and, has also been shown to reduce major histocompatibility complex II (MHCII) on monocytes, collectively resulting in reduced cell activation and antigen presentation [59,63,64]. Signaling through TLR2 has been implicated in facilitating pathogenic mycobacteria’s ability to upregulate IL-10 [44,59]. T cell subsets largely contributing to IL-10 production are CD4+CD25+ regulatory cells, and in the absence of CD25+ cells IFN-γ production is significantly enhanced [65]. Fresh, unstimulated PBMCs from cows with clinical paratuberculosis have been shown to have significantly lower CD25 expression compared to subclinical and control cows but following MPS activation no differences among infection status groups were observed [66]. When T cell subsets were parsed out, MPS stimulated PBMCs from subclinical and clinical cows had significantly higher CD4+CD25+ T cells, and this observation has been replicated [67].

An additional T cell subset found in significant proportions in the bovine is the γδ T cell. This subset can constitute up to 60% of all T cells, and are speculated to bridge responses between innate and adaptive immunity. They can express cytokines such as IL-2, IL-10, IL-12, IL-15, and IFN-γ [68] and have been found to have regulatory function through spontaneous IL-10 secretion which has negative effects on proliferation for CD4+ and CD8+ T cells [69]. In the presence of *Mycobacterium bovis* (*M. bovis*) infected dendritic cells, γδ T cells upregulate production of pro-inflammatory IFN-γ and IL-12 [70]. Additionally, γδ T cells have been reported to be significantly lower in cattle with clinical paratuberculosis [66,71].

Furthermore, investigation of γδ T cell distribution in granulomatous tissue in the bovine has shown significantly higher amount of these cells localized to late stage lesions in naïve calves experimentally infected with MAP when compared to vaccinated calves, which also had higher lesion scores [72]. This experiment begs the question of whether γδ T cells are lower in clinical cows due to them exiting the periphery to assist in controlling bacteria at the site of infection. Granulomatous lesions in paratuberculosis are largely unorganized and resemble that of Type II lepromatous granulomas, with disordered structure and macrophages containing high amounts of bacteria [73,74]. Granuloma formation at the site of infection is dependent upon pro-inflammatory TNF-α expression, and blocking TNF-α expression can result in downregulation of IFN-γ, IL-12, IL-10, IL-17, and nitric oxide production [75]. In *M. bovis* infected cattle, antigen-specific responses by T cells show CD4+ T cells are significant producers of IL-22, while γδ T cells are the main source of IL-17A but can also concurrently produce IL-22 [76]. IL-17A has been associated with early responses in mycobacterial infections [77] and can be produced by CD4+ and CD8+ T cells outside of antigen presenting cell stimulation [78].

## 6. Intracellular Survival and Disease Propagation

### 6.1. Iron Acquisition

One of the most notable features of MAP is its slow growth, which is attributed to its lack of endogenous mycobactin siderophore production. This is in contrast to *Mycobacterium tuberculosis* (*M. tb*), whose growth within host macrophages is dependent upon mycobactin synthesis [79,80]. Sequence analysis of the MAP genome revealed a truncated version of *MbtA*, the first gene in the cluster responsible for mycobactin production [81]. This observation provides a possible explanation for the requirement of exogenous mycobactin in laboratory culture for MAP. Siderophores from unrelated bacterial species have been shown not to support the growth of MAP *in vitro*, while alterations of growth conditions to pH 5.0–6.2 in the presence of bovine transferrin or lactoferrin support MAP growth in culture medium [82]. This study also investigated the presence of mycobactin in ileal tissue from cattle naturally infected with MAP, which ultimately showed no mycobactin was able to be extracted. A genomic island unique to MAP (LSP^P^15) has also been shown to encode genes that contribute to alternative means of iron acquisition [83]. Additionally, MAP infection in a macrophage-epithelial cell co-culture system resulted in MAP gaining iron acquisition capabilities consequential of nitric oxide accrual, further indicating the truncated *MbtA* mycobactin gene is not prohibitive to growth and dissemination through the host [84].

### 6.2. Lysosomal Pathway Maturation

Fusion of endosomal compartments in eukaryotic host cells is a fundamental process involved in recycling of proteins, protein transport, and pathogen degradation. This process is largely mediated by a branch of the Ras superfamily that consists of over 60 small GTPase proteins, which are present in all eukaryotic cells and are highly conserved through evolution due to their essential functions [85,86]. This pathway is a critical function of phagocytic cells of the immune system, such as monocytes, macrophages, neutrophils, and dendritic cells. These cells function in taking up a variety of materials from both the pathogen and host and degrading them.

Mycobacterial regulation of these endosomal trafficking markers has been documented. Firstly, it is likely important to note that during transitions in endosomal markers, a hybrid expression profile of early (Rab5) and late markers (Rab7 and LAMP-1) has been observed in mouse myoblastoma (C2C12) and human alveolar adenocarcinoma (A549) cell lines, suggesting the endosomal maturation pathway may not be totally linear [87].

A proteomics model in sheep has shown that tissue from MAP infected animals can have upregulated expression of Rab5 [88]. Phosphatidylinositol 3-phosphate (PI3P) is an endosomal trafficking receptor that plays a critical role in facilitating fusion of the early endosome and phagosome [89]. ManLAM in the cell wall of *M. tb* can prevent binding and subsequent signaling of PI3P, which has immediate downstream effects through the loss of interaction with early endosome autoantigen (EEA1) that functions to bind the early endosome and tether it to the phagosome, leading to their fusion [90,91,92]. As the majority of Rab5 acquired on the mycobacterial compartment comes from early endosomes, this shows *M. tb* can inhibit maturation steps early in the process and reduce acquisition of Rab5 and its downstream effectors.

Facilitating phagosome-lysosome fusion requires the activity of Rab7-interacting lysosomal protein (RILP) and work in RAW 264.7 cells infected with *M. bovis* BCG show that although Rab7 can be recruited to bacteria-containing phagosomes, acquisition of RILP is impaired and can be a result of Rab7 being bound in an inactive GDP form [93,94,95]. Additional consequences of *M. tb* blocking Rab7-RILP interactions includes reduced lysosomal-associated membrane protein 1 (LAMP-1) recruitment [96]. Impaired Rab7 acquisition may also be explained through excess accumulation of Rab22a, which does not allow for Rab7 recruitment to the compartment containing the mycobacteria [97]. Similar observations have been reported in human THP-1 monocytes infected with live MAP, showing its ability to hinder acidification of the bacteria-containing compartment through preventing RILP binding [98]. Furthermore, the lipid phosphatase SapM produced by *M. tb* directly hydrolyzes and inactivates PI3P [99,100] and a putative SapM protein has been identified in MAP [101].

*M. tb* can also disrupt endosomal trafficking and lysosomal maturation through alternatively splicing of GTPase Rab8B (*RAB8B*) transcripts in human MDMs resulting in ineffective protein [102]. MAP has also been shown to utilize alternative splicing events in bovine paratuberculosis targeting vesicle trafficking, macrophage activation, and lysosomal function [103,104].

A study investigating MAP survival using a J774 murine cell model showed LAMP-1 expression was not significant across 1 hr, 5 hr, and 24 hr timepoints and did not differ between live MAP, dead MAP, and other mycobacterial species investigated. However, LAMP-2 expression was reduced in pathogenic MAP and *Mycobacterium avium* (*M. avium*) compared to killed pathogens and fast-growing *Mycobacterium gordonae* (*M. gordonae*) [105]. Another study using a J774 cells contrasted these results, showing LAMP-1 colocalization with MAP was reduced in cells with live MAP when compared to killed MAP, *Mycobacterium smegmatis* (*M. smegmatis*), and zymosan A [106]. Additional studies have shown LAMP-2 is another late endosomal marker that is downregulated in bovine macrophages infected with MAP [107].

Ultimately, the goal of disrupting normal signaling function of endosomal trafficking markers is to prevent the compartment’s fusion with the acidic lysosome. Intracellular compartments containing live pathogenic mycobacteria, including MAP and *M. avium*, have been shown to maintain mild acidity (pH 6.3) whereas compartments containing these killed bacteria or non-pathogenic *M. smegmatis* and *M. gordonae* can reach a more acidic pH of 5.2 [105]. Exogenous treatment of macrophages with IL-6 and IL-12 cytokines have been shown to upregulate Rab5 and Rab7 expression, respectively [108]. Furthermore, IFN-γ appears to promote endosomal maturation and overcomes mechanisms blocking the pathway. This has been shown by measuring acidification of *M. tb*-containing compartments within macrophages, but this mechanism required sufficient levels of 25(OH)D_3_ [109]. This study also showed IFN-γ induced pro-inflammatory IL-15 production, likely for autocrine activation of the macrophage. A similar IFN-γ mediated mechanism of bacterial destruction is present in monocytes infected with *Coxiella burnetii* [110]. A collective look at regulation of endosomal trafficking markers in mycobacteria shows there are likely species-dependent mechanisms at play, and those mechanisms can be regulated as a result of feedback from the cytokines present in the host cell microenvironment.

### 6.3. Host Lipid Manipulation

Mycobacterial species utilize intracellular cholesterol for survival and persistence within the host [111] through mechanisms including facilitating efficient uptake in macrophages [112] and inhibition of phagolysosome maturation [113]. Macrophages laden with cholesterol are known as foam cells, which provide a microenvironment for *M. tb* that nurtures disease progression in human tuberculosis [114]. MAP manipulates host cellular cholesterol metabolism, with pathogenic cattle strains inducing alterations in efflux and influx pathways creating a competitive environment between the host and bacteria [115]. A study utilizing bovine PBMCs infected with MAP that had been passaged through bovine epithelial cells showed a reduction in activity of cholesterol efflux transporters regulated by the LXR/RXR pathway, including ATP-binding cassette (ABC transporters) ABCA1, ABCG1, and Apolipoprotein E (APOE) [38]. More studies looking at gene expression in epithelial cell processed MAP have also showed upregulation in lipid biosynthesis and metabolism in MAP isolated from secondary epithelial cell cultures, which gain a distinctive pro-inflammatory phenotype as previously discussed [37].

## 7. Vitamin D

### 7.1. History

The story of vitamin D began in the early 1900′s during a time when vitamin deficiencies were more common, and their underlying root cause elusive to physicians. Accessory dietary requirements that were shown to prevent a variety of clinical manifestations led scientists to subscribe to the concept of “vital amines” [116,117,118]. As a result, conventional wisdom of simply balancing dietary proportions of protein, carbohydrates, fats, and salts began to evolve.

Soon after the discovery of Vitamins A, B, and C, McCollum et al. discovered that feeding cod liver, oxidized or not, healed rickets [119]. He also made the observation that developing clinical symptoms of rickets in preparation for disease resolution experiments takes significantly longer in the summer than the winter, one of the first hints that sunlight is important in the mechanism. Work done in the 1930′s confirmed this observation, showing ultraviolet rays convert 7-dehydrocholesterol to vitamin D_3_ in the skin of hogs [120].

### 7.2. Metabolism and Signaling

Two isoforms of vitamin D_3_ are known currently. Vitamin D_2_ is converted from ergocalciferol found in plant material and is known to be a less potent regulator of serum 25-hydroxyvitamin D_3_ (25(OH)D_3_) concentration in humans [121,122]. Animal derived vitamin D_3_ originates from 7-dehydrocholesterol and can be absorbed through the diet or converted in the skin through exposure to ultraviolet rays from sunlight. Further steps in activation of 7-dehydrocholsterol occur through multiple hydroxylation reactions. 25(OH)D_3_, also called previtamin D_3_, is formed in the liver through action of several cytochrome P450 hydroxylases, of which the most common is thought to be CYP27A1 [123,124]. Specifically in cattle, CYP27A1 and CYP2J2 have been associated with regulating incidence of milk fever, which results from dysregulation of calcium homeostasis [125,126]. Next, 25(OH)D_3_ is shuttled through the periphery for local conversion by 1α-hydroxylase (CYP27B1) [127]. Activation of 25(OH)D_3_ by CYP27B1 occurs at a variety of cellular sites, including the kidney [128,129], maternal uterine tissue [130], bone cells [131,132], macrophages [133], and skin cells [134,135]. A visual summary of vitamin D_3_ metabolism is presented in Figure 1.

The half-life of 25(OH)D_3_ is estimated to be around 15 days, allowing for this form to be a reliable indicator of vitamin D_3_ status in the host [136,137]. Comparably, the half-life of 1,25(OH)_2_D_3_ is a fraction of that, estimated at 4–6 h [137]. As a result, concentrations of 25(OH)D_3_ can be found over 1000 times greater compared to 1,25(OH)_2_D_3_ [138]. Additionally, 25(OH)D_3_ is highly stable in serum under proper storage conditions and is largely unaffected by exposure to up to 4 freeze–thaw cycles [139].

Circulation of 25(OH)D_3_ and 1,25(OH)_2_D_3_ in the blood is facilitated largely by vitamin D binding protein (DBP). DBP belongs to the serum albumin protein family. Additionally, a small proportion of 1,25(OH)_2_D_3_ and 25(OH)D_3_ can be transported by serum albumin [140]. Properties of binding affinity to DBP and albumin vary among vitamin D_3_ metabolites, with 25(OH)D_3_ showing the highest affinity but proportions of bound 25(OH)D_3_ and 1,25(OH)_2_D_3_ are similar [140]. However, vitamin D_2_ and its metabolites have a lower affinity for DBP, with an accompanying increased rate of plasma clearance [141]. 25(OH)D_2_ is cleared from the circulation 11 times faster than 25(OH)D_3_, and even greater differences are seen for bioactive 1,25(OH)_2_D_2_ which has been shown to be cleared up to 33 times faster than its D_3_ counterpart [142]. Collectively, these observations are likely attributed to vitamin D_2_′s inferior ability to influence serum 25(OH)D_3_ levels [121]. Execution of biological activity is inhibited by the presence of DBP [143,144,145], so following uptake of the bound molecules by endocytosis, acidification of the compartment facilitates disruption of the DBP-vitamin D bond freeing it for chaperone protein-mediated transport to the mitochondria for activation by CYP27B1 [146]. While considered a minority population, unbound vitamin D_3_ can freely diffuse across cellular membranes [147].

Physiologic activity of 1,25(OH)_2_D_3_ is facilitated through its binding of the nuclear hormone vitamin D receptor (VDR) (Figure 2). Collectively, VDR has been shown to have over 1000 target genes and is found in most tissues [147]. Genomic signaling pathways result in VDR dimerizing with retinoid X receptor (RXR) [148]. This complex directly binds the promoter of genes that possess vitamin D response elements, directly modulating their transcription. In a mouse VDR knockout model, animals observed abnormally increased 1,25(OH)_2_D_3_ serum levels, overexpression of CYP27B1, and nearly undetectable expression of vitamin D_3_-inactiving hydroxylase CYP24A1. Addition of 1,25(OH)_2_D_3_ did not downregulate CYP27B1 or increase CYP24A1 expression in the absence of VDR, indicating that when bound to its ligand VDR helps modulate essential hydroxylase expression [127]. Furthermore, non-genomic signaling of 1,25(OH)_2_D_3_ is thought to be facilitated through the binding of modified membrane VDRs, one of which is called protein disulphide isomerase family A member 3 (PDIA3) [149]. Uptake of the vitamin D_3_ bound PDIA3 complex has been shown to be a result of caveolae-mediated endocytosis [150]. Vitamin D_3_ signaling through these membrane VDRs are thought to modulate rapid responses to 1,25(OH)_2_D_3_ and signaling through PDIA3 can initiate cellular responses through the pro-inflammatory nuclear factor κB (NF-κB) and signal transducer and activator of transcription 3 (STAT3) pathways [151,152].

CYP24A1, the 24-hydroxylase, functions as a regulator of 1,25(OH)_2_D_3_ by hydroxylating the number 24 carbon, which inactivates the molecule and prevents further physiological activity. Additionally, 25(OH)D_3_ is also a substrate for this enzyme and can similarly be inactivated [153]. Ultimately, the newly inactivated metabolites become more polar and water soluble, allowing for excretion in the bile [154,155]. In target cells, CYP24A1 expression and activity is highly induced by increasing amounts of 1,25(OH)_2_D_3_ [156,157,158]. A mouse CYP24A1 knockout model showed dysregulation of 1,25(OH)_2_D_3_ metabolism resulting in excessively high concentrations in the serum [159].

### 7.3. Classical Function

25(OH)D_3_ is considered a prohormone, as it shares similar steroid chemical structure as adrenal and sex hormones [160]. When calcium levels are insufficient, bioactive 1,25(OH)_2_D_3_ facilitates upregulation of calcium transport mechanisms to increase calcium and phosphate absorption by the intestine and renal tubule cells [126]. 1,25(OH)_2_D_3_ facilitates bone growth and remodeling by stimulating differentiation and maturation of osteoblasts and osteoclasts, while remineralization is downregulated by high phosphate and osteopontin levels [161]. Reduced serum calcium concentrations stimulate production of parathyroid hormone from the parathyroid gland. 1,25(OH)_2_D_3_ together with parathyroid hormone also facilitates demineralization of bone to release stored calcium by increasing activity of osteoclasts [162]. When sufficient serum calcium levels have been achieved, calcium inhibits production of parathyroid hormone, which then leads to inhibition of 1,25(OH)_2_D_3_ synthesis by CYP27B1 in the kidney [163]. While these classical functions of vitamin D_3_ are generally well understood in several species, much still stands to be elucidated about vitamin D_3′_s immunomodulatory role, particularly in cattle.

### 7.4. Hydroxylase Expression in Immune Cells

Evidence of 25(OH)D_3_ conversion to 1,25(OH)_2_D_3_ was first shown in macrophages from humans suffering from sarcoidosis [164] and is shown not to be regulated by parathyroid hormone and calcium [165]. Stimulation of 1,25(OH)_2_D_3_ production in monocytes and macrophages is a product of cellular activation through pro-inflammatory cytokines such as IFN-γ, TNF-α, and IL-1β [166,167,168]. Activation of peripheral bovine monocytes by lipopolysaccharide (LPS) induces *CYP27B1* expression, but the concurrent addition of 1,25(OH)_2_D_3_ to cell cultures interrupts expression, bringing it back closer to baseline [169]. Gene expression for both *CYP27B1* and *CYP24A1* have been shown to be reduced by 1,25(OH)_2_D_3_ in bovine PBMCs and MDMs activated by MAP sonicate or live MAP [67,170]. The reduction in vitamin D inactivating hydroxylase in this model may highlight a mechanism that enables monocytes and macrophages to maintain 1,25(OH)_2_D_3_ levels during infection. Cattle with clinical stage paratuberculosis experience abrogated expression of CYP27B1 in the ileum, which may be a feature of inadequate access to its substrate 25(OH)D_3_ as a result of significantly reduced circulating levels in these animals. Interestingly, increased levels of IFN-γ have been associated with upregulation of *CYP27B1* activity [109,171]. Additionally, a *Streptococcus uberis* (*S. uberis*) bovine mastitis model observed localized expression increased for *CYP27B1* in milk CD14+ cells during active mastitis, while CD14- cells saw increased *CYP24A1* [172]. Bovine peripheral blood monocytes treated with 1,25(OH)_2_D_3_ show increased *CYP24A1* transcripts, which are then reduced following activation with LPS [173]. Similar observations have been made in *M. bovis*-BCG vaccinated calves, where secondary purified protein derivative (PPD) antigen exposure abrogated the 1,25(OH)_2_D_3_ induced upregulation of *CYP24A1* [174]. Intramammary treatment of healthy cows with 1,25(OH)_2_D_3_ upregulates expression of *CYP24A1* in total milk somatic cells as early as 4 h post treatment [175], as well as increases *CYP24A1* expression in milk macrophages and neutrophils [173]. Furthermore, the *S. uberis* infected mammary gland overall expressed increased *VDR* as well [172], however VDR expression does not undergo changes in the bovine ileum from MAP infected cows [171,176]. These studies show vitamin D signaling pathways and metabolism may be differentially mediated depending on the offending pathogen.

### 7.5. Host Vitamin D Status and Cathelicidins

Minimum thresholds of 25(OH)D_3_ concentrations reflecting deficiency, insufficiency, and therapeutic immune function have not been concretely established for neither humans nor cattle. An estimation for minimum levels required for immune function is speculated to be 30 ng/mL, whereas baseline concentrations for proper calcium signaling mechanisms is 20 ng/mL [177,178,179,180]. As previously mentioned in the case of bovine paratuberculosis, circulating levels of 25(OH)D_3_ have been shown to be significantly reduced in animals with late stage clinical disease [171]. Work in cattle has shown concentrations of serum 25(OH)D_3_ in adult dairy cows from across the United States at various stages of lactation and supplemented with 30,000–50,000 IU of vitamin D_3_ per day vastly fall between 40–100 ng/mL, with an average of 68 ng/mL [181]. Additionally, seasonality has been shown to impact 25(OH)D_3_ status, with serum concentrations being higher in the summer [182]. This study also reported a high incidence of 25(OH)D_3_ deficiency in calves following birth, observing concentrations mainly between 3–17 ng/mL and over 80% having lower than 20 ng/mL. These data highlight the importance of vitamin D_3_ supplementation beginning from birth, especially during a time of greatest susceptibility to MAP infection.

Efficiency of vitamin D-induced antimicrobial activity is determined by the vitamin D status of the host, as availability of 25(OH)D_3_ substrate is ultimately a limiting factor. Evidence for this observation is reported in human monocytes and macrophages activated by *M. tb* or LPS signaling via TLR2/1 heterodimer or TLR4 binding, where serum with low levels of 25(OH)D_3_ had reduced ability to induce cathelicidin expression [183,184]. Cathelicidins are small cationic peptides that can upregulate expression of chemokines by macrophages, along with increasing phagocytosis of bacteria [185,186]. One functional cathelicidin gene has been identified in humans and mice, while 7 out of 11 cathelicidin genes in cattle have been shown to produce active protein [187,188]. Three of the bovine genes have purported vitamin D response elements, but studies thus far have shown their expression is not induced by 1,25(OH)_2_D_3_ [169]. Other immunologic species differences exist. Cattle are estimated to have over 100 defensin genes while human and mice are estimated to have 39 and 52, respectively [189,190].

Studies on immunologic consequences of low circulating vitamin D levels in cattle are lacking. One related study investigating respiratory syncytial virus (RSV) reported cattle with low serum 25(OH)D_3_ do not see an increase in pro-inflammatory IL-12p40 expression in their lung lesions, whereas animals with high 25(OH)D_3_ observe this benefit [191]. However, severity of the lung lesions was not significantly different between 25(OH)D_3_ status groups. Of additional importance, conventional methods of vitamin D_3_ dietary supplementation in dairy cattle has recently been challenged in a study showing supplementation with 25(OH)D_3_ rather than cholecalciferol may have potential to induce immunologic protective effects against experimentally induced mastitis [192].

### 7.6. Macrophage Phagocytosis and Phenotype

Enhanced capacity for phagocytosis has been observed to be induced in monocytes and macrophages following treatment with 1,25(OH)_2_D_3_ [170,193,194]. In another study, phagocytosis was upregulated in macrophages from healthy individuals that had a low phagocytic index, but those with a current pulmonary tuberculosis infection did not see a benefit [195]. This study used cultured PBMCs in the presence of 1,25(OH)_2_D_3_ for 48 h followed by infection with *M. tb* for 3 h, so in considering the short half-life of 1,25(OH)_2_D_3_ it is possible any 1,25(OH)_2_D_3_-induced effects were not captured for this cell culture method. Small et al. showed increased phagocytosis of *Staphylococcus aureus* and *Candida albicans* with a concurrent upregulation of complement receptor immunoglobulin (CRIg) in human macrophages treated with 1,25(OH)_2_D_3_ [196]. This study utilized complement opsonized microbes; however, CRIg can function as an innate macrophage pattern recognition receptor (PRR) and directly bind some Gram-positive bacteria through recognition of lipoteichoic acids [197]. This posits the question of whether CRIg can also directly recognize any cell wall constituents in mycobacterial species; however, more definitive studies need to be performed to validate the PRR functions of CRIg outside of recognizing opsonized particles.

A study using human monocytes differentiated to macrophages by pro-inflammatory IL-15 showed antimicrobial effects against intracellular *Mycobacterium leprae* (*M. leprae*) were induced by the presence of 25(OH)D_3_ during differentiation [198]. Compared to IL-10 differentiated macrophages, IL-15 differentiated macrophages expressed lower amounts of CD163 and higher amounts of CD209, possibly indicating a more M2-like phenotype. This study did not observe any changes in phagocytosis related to 25(OH)D_3_ treatment, further indicating a M1-like phenotype, as M2 macrophages have been shown to have better phagocytic capabilities [199]. Macrophage phenotype markers CD163 (M1) and CD80 (M2) have recently been shown to have vitamin D_3_-induced changes in expression in bovine MDMs. When infected *in vitro* with MAP, MDMs from cows with subclinical and clinical stage paratuberculosis exhibit reduced CD80 expression following treatment with 1,25(OH)_2_D_3_ [200]. In control cow MDMs infected *in vitro* with MAP, both 1,25(OH)_2_D_3_ and 25(OH)D_3_ reduced CD80. CD163 was largely upregulated by 1,25(OH)_2_D_3_ treatment regardless of activation with MAP. An exception was observed in subclinical cows, which showed 1,25(OH)_2_D_3_ and 25(OH)D_3_ reduced CD163 expression [200]. To fully understand changes elicited by vitamin D_3_, further work including a more expansive macrophage phenotype panel is needed.

### 7.7. Cytokines, Nitric Oxide, and β-Defensins

Modulation of immune responses by exogenous vitamin D_3_ during infectious disease has shown positive outcomes following treatment. PBMCs from *M. bovis* infected cattle treated *in vitro* with 1,25(OH)_2_D_3_ show antigen-specific recall responses through increased nitric oxide production, although pro-inflammatory IFN-γ production was inhibited [201]. A similar reduction in IFN-γ expression has been observed in stimulated PBMCs treated with 1,25(OH)_2_D_3_ from *M. bovis* vaccinated cattle [202]. Nelson et al. also showed a concurrent reduction in gene expression after vitamin D_3_ treatment for IFN-γ, IL-17A, and IL-17F, although only significant for IL-17F during 1,25(OH)_2_D_3_ treatment [174]. Contrasting previous reports, 1,25(OH)_2_D_3_ treatment and activation with MAP has shown to induce a significant increase in *IFNG* transcripts and a concurrent reduction in IFN-γ secretion in PBMC-MDM co-cultures from cattle with naturally acquired paratuberculosis [170]. Similar observations have been reported in PBMCs, with these disparate effects being highlighted in cows in the clinical stage of Johne’s disease [67]. This may highlight a key defense mechanism employed by MAP; however, in another perspective, by reducing IFN-γ production the host may be protected from unnecessary tissue damage while it employs other antimicrobial mechanisms.

Activation of monocytes from healthy cattle by LPS upregulates pro-inflammatory IL-1β (*IL1B*), but the effect is independent of 1,25(OH)_2_D_3_ exposure [169]. In MAP activated PBMC-MDM co-cultures, 1,25(OH)_2_D_3_ significantly upregulates production of *IL1B* transcripts and IL-1β, with an accompanying reduction in *IL10* and IL-10 [170]. However, when PBMCs were cultured alone and activated with MAP sonicate, 1,25(OH)_2_D_3_ increased *IL10* transcripts but reduced IL-10 secretion in cows with paratuberculosis [67]. Similarly activated PBMCs from cows with clinical paratuberculosis observed reduced *IL10* and IL-10 following 25(OH)D_3_ treatment [67]. These data could indicate that vitamin D_3_-induced effects are more efficiently deployed when PBMCs, especially T cells, have the opportunity for crosstalk with macrophages.

In addition to the previously discussed cathelicidin induction, human tuberculosis patients also observe increased nitric oxide production in alveolar macrophages following treatment with vitamin D_3_, but this mechanism is not thought to be regulated by TLR signaling [203]. Human promyelocytic cell line HL-60 has also reported upregulation of nitric oxide production, *NOS2* expression, and a resulting inhibition of intracellular *M. tb* growth following treatment with 1,25(OH)_2_D_3_. 1,25(OH)_2_D_3_ also reduces levels of intracellular *M. tb* in human monocytes and macrophages [204,205]. To further highlight the critical role of vitamin D_3_ in infectious disease, *NOS2* knockout mice infected with *M. bovis* experience increased capacity to kill the bacteria if they have sufficient circulating 25(OH)D_3_ [206]. Work in cattle has shown that dietary supplementation with 25(OH)D_3_ results in a greater proportion of peripheral blood neutrophils with antimicrobial activity, as measured by oxidative burst [207].

1,25(OH)_2_D_3_ has recently been shown to induce nitric oxide production along with upregulating *NOS2* expression in macrophages from healthy cattle that have underwent *in vitro* infection with *M. bovis* [193]. In contrast, MAP activated PBMCs and PBMC-MDM co-cultures from dairy cattle with naturally acquired paratuberculosis experience reduced *NOS2* expression following 1,25(OH)_2_D_3_ treatment [67,170]. Curiously, 1,25(OH)_2_D_3_ significantly upregulated nitrite production, measured as an indicator of iNOS activity, in the activated PBMC-MDM co-cultures from cows with subclinical and clinical paratuberculosis [170]. At this 24 h timepoint, live MAP may target 1,25(OH)_2_D_3_-induced signaling events to disrupt availability of transcripts for protein translation.

In bovine PBMCs activated with MAP sonicate, *CCL5* expression has been shown to be reduced by both 1,25(OH)_2_D_3_ or 25(OH)D_3_ [67]. Other work in peripheral bovine monocytes from healthy dairy cattle has shown upregulated gene expression at 24 h for inducible nitric oxide synthase (iNOS/*NOS2*) and RANTES/*CCL5* in bovine monocytes coordinated by 1,25(OH)_2_D_3_, an effect that is greatly enhanced by concurrent LPS activation [169]. This study also showed *CCL5*, *NOS2*, and nitrite production increase in a dose-dependent manner for 1,25(OH)_2_D_3_ [169]. In a model utilizing PBMCs from calves vaccinated with *M. bovis*-BCG, secondary exposure to *M. bovis* PPD and *in vitro* treatment with 1,25(OH)_2_D_3_ or 25(OH)D_3_ resulted in RANTES/*CCL5* gene expression upregulation [174]. A recent report has similarly shown induction of *NOS2* by 1,25(OH)_2_D_3_ following LPS activation and increased nitrite production [208].

Peripheral blood monocytes from healthy cows treated *in vitro* with 1,25(OH)_2_D_3_ increase expression of various β-defensins, including *DEFB3*, *DEFB6*, *DEFB7*, *DEFB10*, along with *NOS2*, with the effect enhanced further in LPS stimulated cells [173]. Opposite observations were seen in *DEFB5*, whose expression was reduced, and the effect enhanced by LPS. This experiment used an 18 h timepoint, and a follow up experiment using a 4 h timepoint shows peripheral blood monocytes upregulated all bovine β-defensin genes investigated following treatment with 1,25(OH)_2_D_3_, indicating these cellular responses are more robust early in antigen exposure. Recently, studies have shown PBMCs from cattle at different stages of paratuberculosis infection do not have significantly different levels of *DEFB4* and *DEFB7* expression, and there were no notable vitamin D_3_ induced effects [67]. Infection status effects on expression of these β-defensin genes were also not present in co-cultures of PBMCs and MDMs; however, in this model 1,25(OH)_2_D_3_ was shown to significantly reduce *DEFB7* in control and subclinical cows. In milk neutrophils, LPS activation alone at 18 h increased transcripts of *DEFB3*, *DEFB4*, *DEFB7*, and *DEFB10*. The authors further show that 1,25(OH)_2_D_3_ treatment of the mammary gland facilitates upregulation of *DEFB7* at 8 h in milk macrophages [173].

Follow up studies by Merriman et al. show direct treatment of mammary gland with 1,25(OH)_2_D_3_ in healthy cows upregulated expression of *NOS2* and *DEFB7* in total milk somatic cells at 4 h following treatment [175]. When cows with subclinical mastitis had their mammary gland treated with 1,25(OH)_2_D_3_, upregulation was enhanced further and increased expression was observed for *DEFB4* and *DEFB7* along with *NOS2*, with a significant treatment effect being seen by 24 h [175]. 1,25(OH)_2_D_3_ induces localized expression of *NOS2* in milk CD14+ cells and increased *CCL5* in CD14- cells during experimentally induced mastitis caused by *S. uberis* [172]. In a similar study, treatment of mammary gland with 25(OH)D_3_ in cattle with LPS induced mastitis showed total milk somatic cells having upregulated expression of *NOS2*, *RANTES*/*CCL5*, *DEFB3*, *DEFB4*, *DEFB7*, *DEFB10*, *IL1B*, and *IL8*, with most effects being observed between 4–8 h following treatment [209]. Further isolating the source of these responses, the authors showed milk macrophages expressing significantly greater amounts of *NOS2* transcripts when compared to untreated controls, but the significant upregulation of other responses were sourced from neutrophils [209]. Another *S. uberis* induced model of mastitis has shown a resulting decrease in bacterial load in the mammary gland, along with clinical symptoms, after directly treating the gland with 25(OH)D_3_ following each milking [210]. Collectively, these studies may indicate that the type of pathogen and its preferred tissue may heavily influence antimicrobial responses facilitated by vitamin D_3_.

### 7.8. Macrophage Endosomal Trafficking

The desire to understand mechanisms of intracellular MAP survival have driven recent studies to investigate the relationship between vitamin D_3_ and endosomal trafficking markers at different stages of Johne’s disease. 1,25(OH)_2_D_3_ and 25(OH)D_3_ have no significant effects on early endosomal marker Rab5 expression in bovine MDMs; however, control cow MDMs infected with live MAP and treated with either form of vitamin D_3_ experienced significantly reduced Rab5 expression [200]. A notable observation was made in MDMs from cows with subclinical and clinical paratuberculosis, which had significantly reduced Rab5 expression upon *in vitro* MAP infection when compared to MDMs from control cows. Late endosomal marker Rab7 expression was consistently reduced by 1,25(OH)_2_D_3_ treatment in MDMs from control and subclinical cows, regardless of *in vitro* MAP infection [200]. The lack of downregulatory effect in clinical cows posits the question if 1,25(OH)_2_D_3_ treatment has therapeutic potential for these animals, especially considering previous reports of animals in this severe stage of disease having reduced circulating 25(OH)D_3_. Expression of endosomal markers in ileal macrophages from cows in different stages of paratuberculosis have also been recently investigated. Rab5 expression in clinical cow macrophages was shown to be significantly reduced compared to other groups [176]. Total macrophage Rab7 expression was not different among groups, but when colocalized with intracellular MAP subclinical cows showed no detectable association [176]. This may indicate MAP employs virulence mechanisms during early infection to inhibit recruitment of Rab7 to the intracellular compartment it is contained in, which would provide opportunity for MAP’s replication and facilitation of the chronic subclinical phase. Further studies are needed to fully elucidate MAP’s ability to interrupt the phagosomal maturation pathway and the role vitamin D_3_ plays.

## 8. Conclusions

MAP is a stealthy intracellular pathogen that has proven difficult to study, as it possesses several virulence factors that it seems to strategically deploy in response to its immediate environment. Species differences in immune function make model selection important when studying paratuberculosis pathogenesis. This is especially pertinent to the bovine, whose antimycobacterial immune responses do not align with those observed in humans. Specific mechanisms by which MAP hijacks protective immune responses in cattle have yet to be fully elucidated and extensive work is required to understand MAP’s impact on intracellular signaling dynamics of the endosomal trafficking pathway, particularly at different stages of paratuberculosis.

Furthermore, there is a clear ability of vitamin D_3_ to modulate several immune responses in bovine macrophages and PBMCs, especially during MAP infection. This includes regulation of pro-inflammatory mediators IL-1β, IL-12A, IFN-γ, and nitric oxide, along with endosomal trafficking markers Rab5 and Rab7. Increased rates of MAP phagocytosis are also evident in bovine macrophages treated with vitamin D_3_, accompanied by changes in phenotypic markers CD80 and CD163. Additional work is needed to fully understand vitamin D_3_ induced changes in T cell—macrophage crosstalk during infection and how it contributes to resolution of infection both *in vitro* and *in vivo*, as supplementation dose, vitamin D_3_ analogue, and stage of MAP infection may play a significant role.

Our conventional understanding of vitamin D_3_ signaling may expand the bounds of classical VDR binding and interaction with VDRE-induced gene elements. Expression of non-traditional, alternative VDRs at the cell surface may have downstream signaling events that intersect with pathways that impact the host’s ability to deploy antimicrobial tactics such as pro-inflammatory cytokine expression. Further analysis of vitamin D_3_ binding targets, such as membrane PDIA3, and its interactions with downstream signaling pathways is essential to understand the complex dynamic of host responses to pathogenic mycobacterial infection, particularly in cattle, and any implications it may have on clinical stage paratuberculosis where cattle experience largely maladaptive immune responses.

Considerations of the host’s nutritional status on immune function adds another layer of complexity to the host–MAP relationship but is necessary due to the nature of this disease inducing intestinal inflammation. Further work investigating dietary supplementation with 25(OH)D_3_ is needed to evaluate functional outcomes on health metrics such as serum mineral levels and resolution or prevention of naturally acquired infections. Additionally, studies that aim to determine at what *in vivo* 25(OH)D_3_ concentrations immune function suffers from insufficiency or deficiency and benefits from supplementation would help identify therapeutic potential of vitamin D_3_. Applying these data to investigate the impact of vitamin D_3_ status on incidence, along with severity of paratuberculosis is necessary. This knowledge could translate to real-world application for producers to implement intervention strategies to positively impact the health of dairy cattle, ultimately resulting in a more efficiently produced commodity.

## Figures and Tables

**Figure 1 microorganisms-10-01865-f001:**
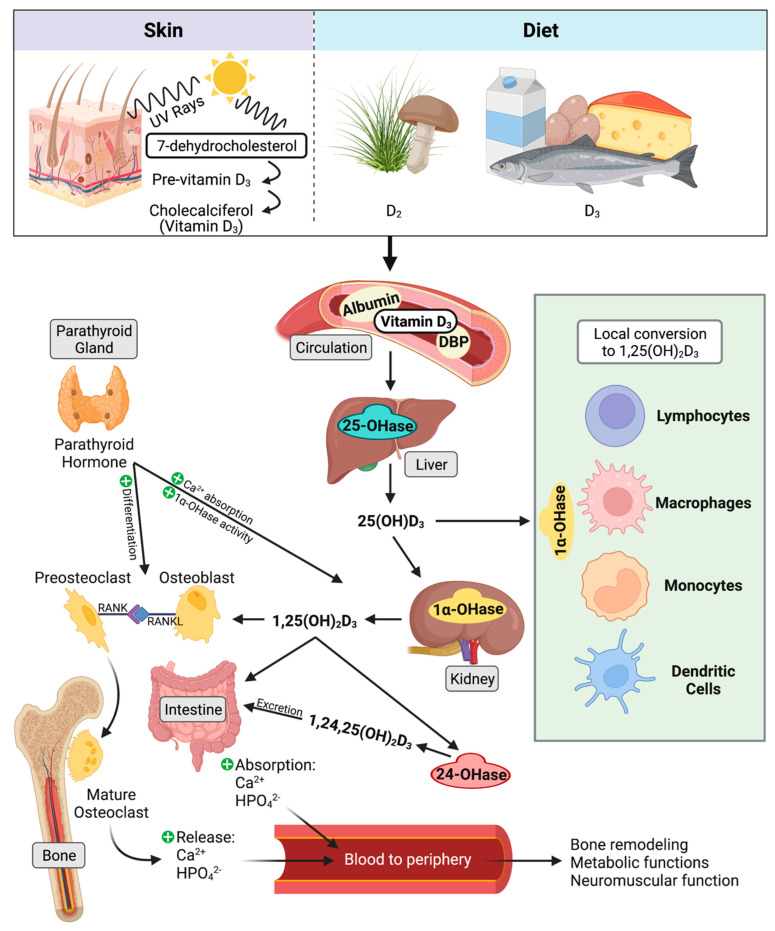
Vitamin D is synthesized in the skin or obtained from the diet. The most common and more potent form, vitamin D_3_, is focused on in this review. Vitamin D_3_ is converted to 25(OH)D_3_ in the liver. Classical effects on calcium homeostasis begin in the kidney, where 25(OH)D_3_ is converted to bioactive 1,25(OH)_2_D_3_. 1,25(OH)_2_D_3_ increases osteoclast differentiation and activation, and upregulates calcium and phosphate absorption in the gut. Concurrently with 1,25(OH)_2_D_3_, parathyroid hormone stimulates osteoclasts. It additionally upregulates 1α–OHase (CYP27B1) activity and calcium reabsorption in the kidney. 25(OH)D_3_ and 1,25(OH)_2_D_3_ are both inactivated by 24–OHase (CYP24A1) and are excreted in the bile. Non-classical signaling of vitamin D_3_ involves local activation of 25(OH)D_3_ to 1,25(OH)_2_D_3_ by 1α–OHase (CYP27B1) in cells of the immune system. Figure created using BioRender.com (accessed on 28 July 2022).

**Figure 2 microorganisms-10-01865-f002:**
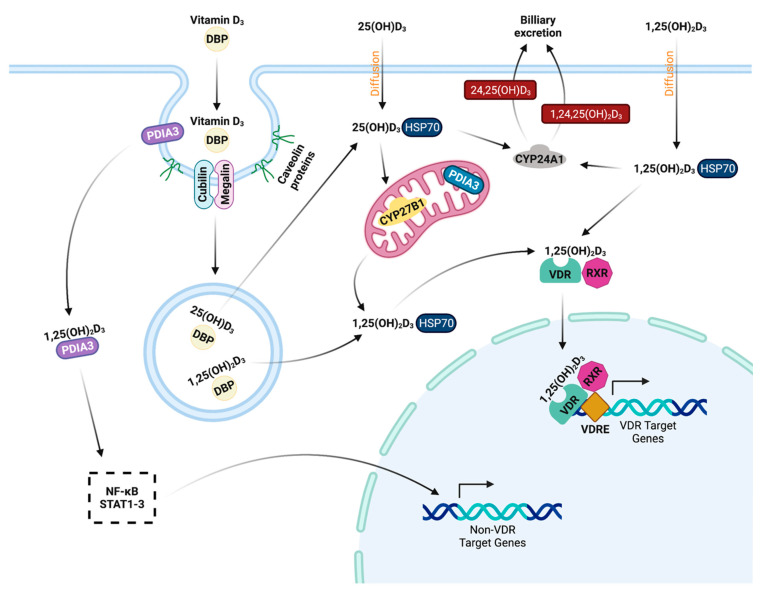
25(OH)D_3_ and 1,25(OH)_2_D_3_ travel in the circulation mainly bound to vitamin D binding protein (DBP). They are taken up by caveolae mediated endocytosis, where DBP then disassociates. Free 25(OH)D_3_ and 1,25(OH)_2_D_3_ are lipophilic and can diffuse across the cell membrane. 25(OH)D_3_ is activated by 1α-OHase (CYP27B1) in the mitochondria. Activated 1,25(OH)_2_D_3_ binds the vitamin D receptor (VDR) and forms a complex with retinoid X receptor (RXR). Together, they bind vitamin D response elements (VDREs) in vitamin D target gene promoters to facilitate gene expression. Additionally, alternate receptors at the cell membrane have been shown to bind 1,25(OH)_2_D_3_. One such receptor, protein disulphide isomerase family A member 3 (PDIA3) can bind 1,25(OH)_2_D_3_ and interact with NF-κB and STAT1–3 pathways, showing 1,25(OH)_2_D_3_ can also indirectly influence gene expression outside of VDR target genes. PDIA3 is also expressed in the mitochondria, but its full role is currently unknown. Figure created using BioRender.com (accessed on 28 July 2022).

## Data Availability

Not applicable.

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
