# Peer review of "Bovine Immunity and Vitamin D3: An Emerging Association in Johne’s Disease"

_microorganisms, 2022, doi:10.3390/microorganisms10091865_

Round 1
Reviewer 1 Report
As could be observed, the purpose of this work is to provide a revision of different reports on the bovine immune system that in summary pinpoints the association of the biological effect and a deficiency of D vitamin with an M. avium infection in cattle.
The first questions authors need to address are who is this review for, what prior knowledge is required to understand the review, and what is the core message being conveyed.
In general, the main body of information is focused on the immunological aspects of MAP infection in cattle, afterward, only a single and extensive subtheme is dedicated to the physiology and biological aspects of D vitamin and to independent works that performed experimental basic studies from homeostasis of D vitamin to gene expression in different biological models that include cellular and animal including human beings, even those in other infectious diseases.
Thus, it would be desirable to separate and rewrite 2 more sections to improve comprehension.
Most references are not too recent which is understandable given the paucity of data on the relation of deficiencies of D vitamin with mycobacterial disease outcome, is recommended to change/omit the proportion of actualized references vs previous works, shortening and summarising those new sections.
The concluding section and outlook need to be substantially improved with personal thoughts and if possible by including an additional figure of hypothetical effect on immunology within MAP fate of infection and suggest aspects or research topics.
Reviewer 2 Report
Please find attached.

Round 2
Reviewer 1 Report
The manuscript was improved as requested and the overall content is better organized and presented.
Although the final section with personal comments was improved the authors still do not precisely reflect what are those gaps in the field of immunology of mycobacterial infections and vitamin D to be considered as a new line of study and to open a whole research field itself.
It would require to address in this final section who are the best receptors of such information and which are the main benefits for the cattle industry.
